# Odor Profile of Four Cultivated and Freeze-Dried Edible Mushrooms by Using Sensory Panel, Electronic Nose and GC-MS

**DOI:** 10.3390/jof8090953

**Published:** 2022-09-11

**Authors:** Inmaculada Gómez, Rebeca Lavega González, Eva Tejedor-Calvo, Margarita Pérez Clavijo, Jaime Carrasco

**Affiliations:** 1Department of Biotechnology and Food Science, University of Burgos, Plaza Misael Bañuelos s/n, 09001 Burgos, Spain; 2Centro Tecnológico de Investigación del Champiñón de La Rioja (CTICH), Carretera Calahorra, KM 4, 26560 Autol, Spain; 3Centro de Investigación y Tecnología Agroalimentaria de Aragón (CITA), Instituto Agroalimentario de Aragón—IA2 (CITA-Universidad de Zaragoza), Avda. Montañana 930, 50059 Zaragoza, Spain; 4Department of Production and Characterization of Novel Foods, Institute of Food Science Research—CIAL (UAM + CSIC), C/Nicolas Cabrera 9, Campus de Cantoblanco, Universidad Autónoma de Madrid, 28049 Madrid, Spain; 5Department of Biology, University of Oxford, South Parks Road, Oxford OX1 3RB, UK

**Keywords:** edible mushrooms, odor profile, electronic nose, sensory analysis, volatile compounds

## Abstract

Cultivated mushrooms are well-known nutrient inputs for an equilibrated diet. Some species are broadly appreciated due to their medicinal properties. Lately, a number of novel foods and nutraceuticals based on dehydrated and freeze-dried powder obtained from cultivated mushrooms has been reaching the market. The food industry requires fast and reliable tools to prevent fraud. In this, work we have cultivated *Agaricus bisporus* sp. *bisporus* (AB) (white button mushroom), *Agaricus bisporus* sp. *brunnescens* (ABP) (portobello), *Lentinula edodes* (LE) (shiitake) and *Grifola frondosa* (GF) (maitake) using tailor-made substrates for the different species and standardized cropping conditions, which were individually freeze-dried to obtain the samples under evaluation. The aim of this article was to validate the use of two different methodologies, namely, electronic nose and sensory panel, to discriminate the olfactory profile of nutritional products based on freeze-dried mushrooms from the different cultivated species. Additionally, GC-MS was used to detect and quantify the most abundant volatile organic compounds (VOCs) in the samples. The multivariate analysis performed proved the utility of electronic nose as an analytical tool, which was similar to the classical sensory panel but faster in distinguishing among the different species, with one limitation it being unable to differentiate between the same species. GC-MS analysis showed the chemical volatile formulation of the samples, also showing significant differences between different samples but high similarities between varieties of the same cultivated species. The techniques employed can be used to prevent fraud and have the potential to evaluate further medicinal mushroom species and build solid and trustful connections between these novel food products and potential consumers.

## 1. Introduction

The food supply chain requires customer trust to generate loyalty towards novel products. Lately, consumers are aware of the negative impact of a diet largely based on meat, due to environmental impacts, animal welfare or health issues, ending up in becoming reluctant to eat meat-based products but also to novel alternative foods [1]. Cultivated mushrooms have been consumed and appreciated for centuries in Asia [2]; however, they can be considered a novel food in Europe, where consumers are showing an increasing interest in alternative protein food products, and therefore an increasing demand for mushroom is projected [3].

Although many different species are harvested from the wild, since it is estimated that around 700 mushroom species are edible [4], only a few, 130, have been domesticated [2], and just around 12 species are cultivated for commercial use as food or medicine [5].

Some species are appreciated due to their broad-spectrum medicinal and pharmacological properties. Among the health benefits reported, cultivated medicinal mushrooms exhibit antibiotic, cytotoxic, immunomodulating, anti-inflammatory, antioxidative, antiallergic, antidepressive, antihyperlipidemic, antidiabetic, digestive, hepatoprotective, neuroprotective, nephroprotective, osteoprotective, and hypotensive activities [6]. Many different bioactive metabolites have been extracted and characterized from cultivated medicinal mushrooms, including carbohydrates, sterols, sphingolipids, fatty acids, sesquiterpenes, peptides, or enzymes [7]. For instance, cultivated edible and medicinal mushrooms, such as *Agaricus bisporus* (button mushroom) and *Lentinula edodes* (shiitake), stand out for their high content in β-glucans [8]. Antioxidant substances/free-radical scavengers, such as ascorbic acid, caffeic acid, adenosine, ergosterol, ergothioneine or glutathione, have been detected in the fruiting body and mycelium of *Hericium erinaceus* (lion’s mane) [9]. Furthermore, mushroom bioactive peptides (MBAPs), which have antioxidant and antimicrobial activities, and can effectively reduce blood pressure, have been reported in *Agaricus bisporus*, *Grifola frondosa* (hen of the woods) or *Letinula edodes* [10].

Gas chromatography–mass spectrometry (GC-MS) combines gas chromatography (for separation of compounds) and mass spectrometry (for the structural determination of compounds) to identify different substances within a volatile test sample [11]. Noteworthy, the odor profile of the cultivated genera *Agaricus* (showing a high content of alcohols and aldehydes) [12], *Lentinula* (with sulfur-containing compounds, eight-carbon compounds and aldehyde compounds, greatly contributing to mushroom flavor) [13] and *Grifola* (showing esters, alkanes, aldehydes and terpenes as major components) [14] has been analyzed by GC-MS. However, the mechanisms driving the odor profile of mushrooms by the substrate used or the drying process applied is not well described nor understood [12,13].

In addition to the GC-MS, the odor profile of fresh and dried samples of cultivated button mushroom (*Agaricus bisporus*) has been evaluated by using an electronic nose, with comparable results [12]. The electric nose is an analytical device that permits automatic discrimination among different samples based on the odor profile (composition of their volatile fraction). In this device, various sensors exhibit a particular selectivity and sensitivity with respect to singular components of the sample, eventually generating a “fingerprint” of the chemical image in the volatile mixture [15].

Although analytical devices can prevent subjective mistakes, humans are able to detect a broad range of odors since the human olfactory epithelium contains ≈10^6^–10^7^ total olfactory receptor neurons [16]. Triangle sensory tests are discriminatory tests that aim to establish differences among diverse samples instead of evaluating subjective sensation experienced by the tested individuals [17]. Triangular tests have been successfully employed for discrimination of samples from wild edible and toxic mushroom of the *Amanita* genus, in combination with an electronic nose [18]. Many different types of mushrooms are offered as functional foods and source of nutraceuticals [19]. Those products can be found in different commercial formats, including dehydrated mushroom powder in the form of pills or tablets, often commercialized as a probiotic, such as “Bio Shiitake” (Hifas da Terra, Spain) from shiitake powder; purified extracts obtained from the fruiting bodies or fermented mycelium, usually commercialized as natural nutritional complements with concentrated active ingredients, such as “Mico Shii”, a pure extract from shiitake (Hifas da Terra, Spain); or a mixture of both a mushroom extract and mushroom powder, such as “Maitake 180” (Hifas da Terra, Spain).

The cultivation of medicinal mushrooms is expensive due to the reduced biological efficiency of the substrates (kg of fresh mushrooms harvested in 100 kg of dried substrate) and profitability relies upon ensuring high prices for producers. Due to the novelty of these products, there are few reliable tools to differentiate among mushroom powder that can be produced out of commercial low-priced species globally cultivated, including button mushroom, which represents 15% of the global production [20], and the most commonly cultivated in Europe [20,21].

The relative content of volatile compounds among cultivated mushrooms following standardized procedures—harvested and freeze-dried to powder—for potential use as nutraceuticals or food supplement, has not been previously compared. This work aims to validate the use of different methodologies, namely, electronic nose, sensory panel and GC-MS, to elucidate the odor profile and discriminate among nutritional products based on freeze-dried powders obtained from the following cultivated species of edible and medicinal mushrooms: *Agaricus bisporus* sp. *bisporus* (AB), *Agaricus bisporus* sp. *brunnescens* (ABP); *Lentinula edodes* (LE) and *Grifola frondosa* (GF).

## 2. Materials and Methods

### 2.1. Cultivated Mushrooms under Study

Mushrooms are heterotrophic organisms that require selective substrates and specific environmental conditions to produce commercially viable crops. The mushroom species selected in this study were *Agaricus bisporus* sp. *bisporus* (AB) (white button mushroom), *Agaricus bisporus* sp. *brunnescens* (ABP) (portobello), *Lentinula edodes* (LE) (shiitake) and *Grifola Frondosa* (GF) (maitake).

Both species of *Agaricus* were cultivated following standardized protocols [22] in the growth chambers available at the Mushroom Technological Research Center (CTICH) (Spain) equipped with a device for climate control and hermetic doors to facilitate optimal conditions for mushroom cropping and preventing pests and diseases. Phase II substrate spawn with 1% Amycel^®^ XXX (white variety) and Amycel^®^ Heirloom (Portobello variety) (Amycel^®^-Spawn Mate^®^, Watsonville, CA, USA) were used.

Shiitake (*Lentinula edodes* strain ST1 from Fungisem, Spain) and maitake (*Grifola frondosa* strain M9827 from Mycelia, Belgium) were cultivated on sterile substrates based on wood chips, cereal bran, cereal straw, carbonate, corn, oat and a commercial supplement, following standardized protocols [23]. The humidified mix substrate was introduced in polyethylene plastic bags (HDPE), with a Tyvek^®^-type filter on the upper side to facilitate gas exchange and then autoclaved at 121 °C for 4–5 h. After cooling, the sterile substrates were inoculated with the cited commercial spawn and incubated until complete colonization at 25 °C in the dark in the incubation rooms available at CTICH, Spain. The colonized substrates were taken to the cultivation room where fructification was induced.

The four mushroom species were harvested following commercial requirements (close caps in the case of *Agaricus*) for fresh mushrooms, at the mature stage. Healthy mushrooms were selected, cut and deep frozen in an ultra-freezer at −80 °C. Subsequently, the mushrooms were freeze-dried and grinded into a powder (Appendix A).

### 2.2. Color

In order to compare the color among the samples under study, spectrophotometric analysis was performed.

Instrumental measurement of the color of the freeze-dried mushrooms was performed employing a Konica Minolta CM-2600d spectrophotometer (Konica Minolta Business Technologies Inc., Tokyo, Japan). A D65 illuminant and 10° standard observer were selected. The color coordinates were determined in the CIELAB color space, expressing the results in terms of L* (lightness; L* is represented on the vertical axis of the color space diagram, with values from 0 (black) to 100 (white)), b* (yellowness; the b* value indicates the yellow-blue component of a color, where positive and negative values indicate yellow and blue, respectively) and a* (redness; the a* value indicates the red-green component of a color, where positive and negative values indicate red and green, respectively) [24]. Mushroom powder color was measured in quintuplicate—five different technical replicates from the same biological sample.

### 2.3. Sensory Analysis—Triangle Sensory Test

The sensory analysis was carried out by a group of 54 panelists located in the Science Faculty (University of Burgos, Spain) facilities, following the recommendations for tasting-room conditions [25]. Panelists read the informed consent form for sensory evaluation. A triangle test of freeze-dried mushroom samples based on odor was performed according to ISO 4120 [26].

The samples were presented in equal quantities (0.25 g), in opaque and odorless glass vials, thus preventing panelists from recognizing the mushrooms by any other sense than smell. Likewise, tests were conducted under red light in order to mask even a slight color difference. Samples were randomly coded with three digits. Each assessor was instructed to open the cover of the cup, to inspire the headspace deeply and then to close it before opening the next one. Their aim was to determine which of the three sampled products was perceived as different from the other two by sniffing only.

The samples were evaluated in a randomized complete block design. Thus, the panelists had to analyze six different series (Table 1). The test was developed in two sessions on different days, three series per session. After smelling the samples of a series for an interval of time from 30 to 60 s, the panelists proceeded to select a different sample. In order to prevent the olfactory receptors from reaching saturation, a waiting time of 15 min was employed between series. No information was given to the panelists about the origin of the samples.

### 2.4. Odor Profile—Electronic Nose

Odor profile was measured directly on the grounded, freeze-dried mushroom samples. Samples were measured ten times for each type of freeze-dried mushroom. The methodology followed was the one described by García-Lomillo et al. [27], with some modifications. The extraction of volatile compounds was performed by static head space. In short, 0.25 g of freeze-dried mushroom were introduced in vials and incubated during 10 min at 55 °C with an autosampler (HS100 CTC-Combi-Pal, CTC Analytics AG, Zwingen, Switzerland) to determine the odor profile of each type of mushroom powder. The extracted volatile fraction (a volume of 1.5 mL of the head space) was injected into the Alpha MOS electronic nose, model α-FOX 4000 (AlfaMOS, Toulouse, France), equipped with metal oxide sensors, and available at the Food Technology Area of Burgos University. The main task of this device is automatic discrimination of the samples based on differences in the composition of their volatile fraction (odor profile). Analysis of the sensor responses collected for two minutes was performed. The results of the electronic nose sensors obtained (18 different sensors) were processed obtaining the fingerprints of the volatile organic compounds (VOCs) present in each analyzed mushroom. The resistance in each sensor was assayed by the software AlphaSoft version 9.1 and the response intensity was calculated as previously described [27].

### 2.5. Identification and Quantification of VOCs, GC-MS

Samples were analyzed three times for each type of freeze-dried mushroom.

#### 2.5.1. Extraction by Solid-Phase Microextraction (SPME)

The methodological approach was based on works carried out by Tejedor-Calvo et al. [28], with some modifications. For that, a fused silica fiber coated with a 50/30 mm layer of divinylbenzene/carboxen/polydimethylsiloxane from Supelco (Barcelona, Spain) was chosen. The samples (2 g of powder material) were placed in a 20 mL glass vial closed with a septum. After the vial was conditioned at 50 °C for 10 min, the fiber was exposed to the headspace of the vial for 20 min.

#### 2.5.2. GC-MS Analysis

The VOCs profile of the different mushroom species was analyzed by a static Agilent 6890N system GC-MS-FID (Termoquest, Milan, Italy). For that, freeze-dried samples (2 g) were placed in 20 mL vials and hermetically closed. This instrument was equipped with a capillary column (HP-5MS, Agilent Technologies, Santa Clara, CA, USA) of 30 m, 0.32 mm i.d., 0.25 μm film thickness and a flow of 1 mL/min, with helium as a carrier gas. The oven temperature was 45 °C held for 2 min, 45–200 °C at a rate of 4 °C/min, and finally to 225 °C at 10 °C/min, and held for 5 min. The MS used the electron impact mode with an ionization potential of 70 eV and an ion source temperature of 200 °C. The interface temperature was 220 °C. The MS scanning was recorded in full scan mode (35–250 *m*/*z*). TurboMass software was used for controlling the GC-MS system.

Peak identification of the VOCs was achieved by comparison of the mass spectra with mass spectral data from the NIST MS Search Program 2.0 library (US National Institutes of Standards and Technology, Gaithersburg, MD, USA), and by comparison of the previously reported retention index (RI), with those calculated using an n-alkane series (C6–C20) under the same analysis conditions. The n-alkanes series and standards for MS identification (all standards of purity higher than 95%) were purchased from Sigma-Aldrich (Madrid, Spain). Semiquantification was done by integrating the area of one ion characteristic of each compound and normalization by calculating the relative percentage. This allowed comparison of each eluted compound between samples.

### 2.6. Statistical Analysis

Color instrumental results were analyzed by one-way analysis of variance (ANOVA), and statistical comparisons of the different mushrooms were performed using an LSD Fisher’s test (*p* ≤ 0.05).

Results of the sensory evaluation were analyzed according to ISO 4120:2021 [26]. Values of α = 0.05, β = 0.10 and pd = 30% were adopted, aiming at ensuring, within a 95% confidence level, that no more than 30% of the population can detect a difference among the samples.

For the statistical treatment of the volatile compounds and the odor profile by the electronic nose, an algorithm was used to calculate the principal components among the samples. This method assumes collinearity between the variables involved. The heatmap plot and multivariate analysis (principal component analysis (PCA)) were performed and visualized in Statgraphics Centurion XIX software (Statpoint Technologies, Warrenton, VA, USA).

## 3. Results

### 3.1. Color

The color of pulverized freeze-dried mushrooms is not well-known among consumers, even when these differ among cultivated species. In order to evaluate differences in color among the species under study, the instrumental color of the powders was measured. Statistically significant differences in the parameters L*, b* and a* were measured (Table 2), discriminating by species of cultivated mushrooms.

AB and ABP did not differ in the L* value; however, the powder from both *Agaricales* species under evaluation were significantly brighter than LE and GF. Yellowness of the four powders, measured by the b* value, differed significantly, with the highest value for LE and the lowest value registered for ABP. Ultimately, redness, recorded by the a* value, showed significant differences among the four samples, with GF showing the highest value (more red) and LE the lowest one (less red).

### 3.2. Sensory Analysis—Triangle Sensory Test

It is worthwhile to note that, according to the statistical table (as reported in [26]) comprising 54 panelists, the number of correct responses should be at least 25, so that the sensory panel is able to detect a difference between two samples. In the present experiment with the sensory panel, the number of correct responses (Table 3) for five out of six series was enough to conclude that there were significant differences between freeze-dried mushrooms. However, no statistical differences were found for AB and ABP, which are two commercial varieties from the same species, *Agaricus bisporus*.

### 3.3. Odor Profile—Electronic Nose

Differences were found in the olfactory profile, discriminating by species of cultivated mushrooms.

The sensor intensity collected showed similarities in between AB and ABP, while GF and LE clearly differed (Figure 1). In the analysis of principal components, while plotting the response intensity of the 18 different sensors (Figure 2), two of the components accounted for 98.66% of the variability in the data. The sensors P10/2 (PC1: −0.25; PC2: −0.06), T70/2 (PC1: −0.25; PC2: −0.06), P30/2 (PC1: −0.03; PC2: 0.62) and T40/1 (PC1: −0.25; PC2: −0.06) had a significantly different response, depending on the species (Figure 1 and Figure 2).

### 3.4. Identification and Quantification of VOCs by GC-MS

The quality of the data gathered was tested by checking for a coefficient of variation <1%, which was detected in most of the compounds. Appendix A compiles the percentage of the most-abundant VOCs detected in the freeze-dried samples by GC-MS analysis.

As noted in the heatmap (Figure 3), the most abundant VOCs in the samples of white button mushroom and Portobello mushroom were the closely related pair 1-hexenol (23.33% in AB and 27.75% in ABP) and hexanal (11.42% in AB and 12.41% in ABP), which are described as a fruity aroma [29]; the closely related pair 3-octanone (14.72% in AB and 10.99% in ABP) and 1-octen-3-ol (6.76% in AB and 6.30% in ABP), secondary metabolites present in most mushrooms and responsible of their typical “mushroom-like” odor [30]; and benzaldehyde (11.96% in AB and 9.34% in ABP), described as one of the most dominant compounds in raw champignon [31].

In shiitake, the most abundant compound detected was butanal-3-methyl (13.85%), an aldehyde described as ethereal, chocolate, peach and fatty, which is also detected in black Perigord truffle [32]; this was followed by hexanal (11.51%) and 1-hexanol (7.93%). Ultimately, maitake showed a higher content of 3-octanone (16.25%) and hexanal (15.63%), followed by 3-octanol (6.73%) and butanal-3-methyl (6.17%) (Figure 3).

Regarding Figure 4, statistically significant differences were found among the odor profiles obtained by GC-MS from the freeze-dried products.

In the analysis of the principal components, plotting the most-abundant compounds detected by GC-MS, two of the components accounted for 93.19% of the variability in the data. As detailed in Figure 4, the species AB (PC1: −2.79; PC2: −1.79) and ABP (PC1: −2.21; PC2: −1.96) cluster together, showing low variability in their odor profile, but significantly differ in the first component with LE (PC1: 6.65; PC2: −0.38) and in the second component with GF (PC1: −1.65; PC2: 4.12). Both LE and GF had a very different odor profile, showing significantly different responses in both components of the analysis (Figure 4).

## 4. Discussion

Even when we have shown differences in the color of the samples under evaluation, consumers are not familiar with freeze-dried mushroom powder and therefore we cannot expect a subjective response identifying species and even the color could be easily subjected to manipulation with artificial color additives [33].

Discriminatory sensory tests establish the differences that may exist between two or more samples [17]. In this study, a triangle sensory test was performed. No research has been found in the literature in which a sensory analysis was used to evaluate the similarity among freeze-dried powder obtained from cultivated mushrooms. However, Portalo-Calero et al. [18] used the triangle test to evaluate Amanita species and reported that humans are capable of differentiating between mushrooms through smell. In this way, the results could be compared to those obtained from the tests with the electronic nose. It should be highlighted that, in our study, the panel was not able to discriminate between two varieties of the same species, Agaricus bisporus (AB and ABP), although the three different species under evaluation were differentiated by triangle test. Similarly, the triangle test, by using a sensory panel and electronic nose, was able to discriminate among different species of the Amanita genus [18], including wild edible and poisonous species, suggesting that the triangle test is limited in discriminating between varieties of the same species but permits differentiation among different species belonging to the same genera. The electronic nose also has been probed to discriminate between the odor profile among different cultivated fresh mushrooms, including golden needle (Flammulina velutipes), white mushroom (Agaricus bisporus), shiitake (Lentinus edodes), eryngii (Pleurotus eryngii), hen of the woods (Grifola frondosa), and shimeji (Hypsizygus marmoreus) [34], which encourage future research to target the potential of the technique using a trapping system to reduce sensor value variation. Similar to the results obtained, e-nose sensors have shown the ability to accurately respond to eight different dried commercial mushrooms, namely, Pleurotus abalonus, Agrocybe aegirit, Hericium erinaceus, Grifola frondosa, Coprinus comatus, Boletus edulis, Lentinula edodes and Pleurotus eryngii, which, according to the authors, suggests the utility of an electronic nose as a tool for the identification of edible fungi [35]. The results suggest that the odor profile of different varieties from the same mushroom species are closely related while it differs for different species of cultivated Basidiomycetes. Noteworthy, the substrates used as a nutrient reservoir for cultivation of both Agaricus species were pasteurized compost and commercial casing [36,37] while LE and GF were cultivated in the same mixture of sterile lignin-rich substrates [38,39], indicating that the species under evaluation build a singular odor profile that does not depend on the substrate formulation. The substrate formulation has significant impact on mushroom parameters, such as biological efficiency, crop earliness, unitary weight, mushroom composition, commercial class and firmness [38,39,40,41]. Likewise, a brief number of reports have evaluated the effect of substrate on the organoleptic characteristics of cultivated mushrooms [13,42]. The relationship between volatile compounds production and culture substrates during Lentinula edodes grown in different substrates formulations was evaluated [13]. The authors conclude that the carbon source for shiitake cultivation can be partially modified by using bagasse instead of sawdust. However, the mechanisms driving the volatile composition as a result of the carbon source used for substrate formulation require further research [13]. Furthermore, variation in the umami taste and aroma (considered an important factor for consumer choice) in cultivated mushrooms also have been correlated to the use of different substrates [42]. However, as noted by the authors, the mechanisms underlying the variations in umami components of different mushrooms are unclear. Therefore, to date, the impact of substrate formulation, as an agronomical trait configuring the odor or flavor profile, is not well understood, and can be the object of future research.

Our chromatographic and spectrophotometric analysis of the freeze-dried, cultivated mushrooms reinforced the discriminative odor profile among the samples under evaluation. When coupled with GC, MS provides a quick scan rate and significant information for all the detectable chemical volatiles in food samples, confirming the potential of this technique to prevent food fraud [43]. The multivariate analysis of the relative content of most-abundant VOCs in the samples quantified by GC-MS showed a different odor profile for the three species AB, LE and GF, although it could not discriminate between the two varieties of Agaricus, AB and ABP. These results are in agreement with those reported by Šiškovič et al. [44], who detected a truffle VOC profile significantly modified compared to fresh and freeze-dried truffles, although the observed VOCs’ transformation during freeze-drying was species specific. However, in their study, the same species collected from different locations in Slovenia did not show very significant differences, suggesting that GC-MS may have limitations in differentiating between the same species, which is also reflected in our results. Future studies should also target the VOC profiles of freshly cultivated mushrooms and the impact of freeze-drying in altering this profile; still, working with freeze-dried samples prevents sample deterioration and VOC profile alteration due to microbiological activity and fresh mushroom decay. GC-MS has proven to be a reliable technique to characterize the odor profile of eight dried commercial mushrooms, including GF and LE [35]. In contrast to the results obtained by these authors, 1-octen-ol was not the most-abundant alcohol detected in the freeze-dried samples of LE, but 1-hexenol, which was not detected in this previous research; but, as in our research, a high content 1-octen-ol was reported for GF. We have also detected a relatively high content of different aldehydes in the samples of LE and GF, marked by a high content in hexanal and butanal-3-methyl. The different observations reported suggest that oven drying in comparison to freeze-drying can provide different, distinctive volatile profiles of the dried mushroom samples obtained after both procedures. This hypothesis is also reinforced by the results of Du et al. [45], who reported significant sensory differences among three different strains of *A. bisporus*, made sharper by the roasting process. Lentinula edodes (shiitake) and Grifola frondosa (maitake) have been deeply analyzed for their medicinal value. Among their active ingredients, which have multiple beneficial uses, for instance the polysaccharide lentinan from LE [46] and grifon from GF [47], there are immunomodulating and antitumor effects. They require specific sterile substrates and singular conditions for their growth. In comparison to the commonly cultivated button mushroom, the biological efficiency of medicinal mushrooms is significantly lower and therefore specific tools must be designed for the food industry to prevent fraud. The odor profile as characterized by the triangle test, electronic nose and GC-MS have permitted to differentiate among the edible and medicinal mushrooms under evaluation and the results achieved suggest that this research could be extended to other cultivated or even wild species.

## 5. Conclusions

According to the results obtained, an electronic nose has analytical utility and could be used for the evaluation of different types of freeze-dried mushrooms, similar to the classical sensory triangle test, but with greater speed. Furthermore, MS coupled to GC permits to detect, separate, identify and quantify the relative content of the most-abundant volatiles in freeze-dried mushrooms. Our results indicate that the techniques employed can differentiate among freeze-dried samples of commonly cultivated button mushroom and medicinal species such as maitake or shiitake, and therefore could be used to prevent fraud in novel food and nutraceutical products based on cultivated mushrooms.

## Figures and Tables

**Figure 1 jof-08-00953-f001:**
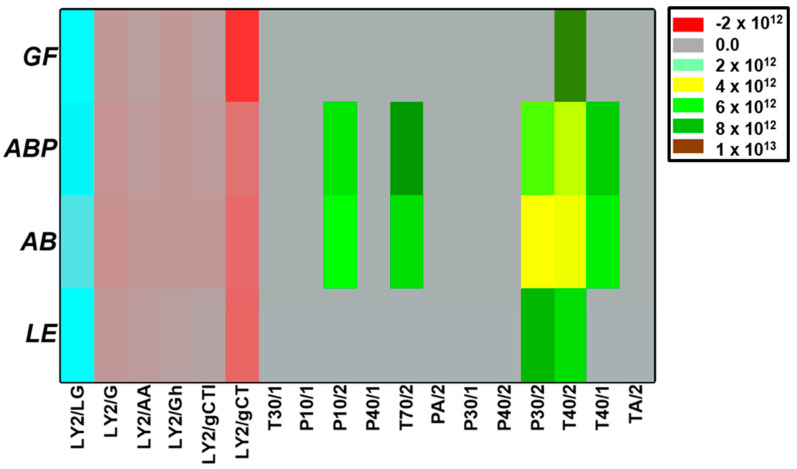
Response intensity of the 18 sensors from the electronic nose for the different freeze-dried samples from cultivated mushrooms (AB: *Agaricus bisporus* sp. *bisporus*; ABP: *Agaricus bisporus* sp. *brunnescens*; LE: *Lentinula edodes*; GF: *Grifola frondosa*).

**Figure 2 jof-08-00953-f002:**
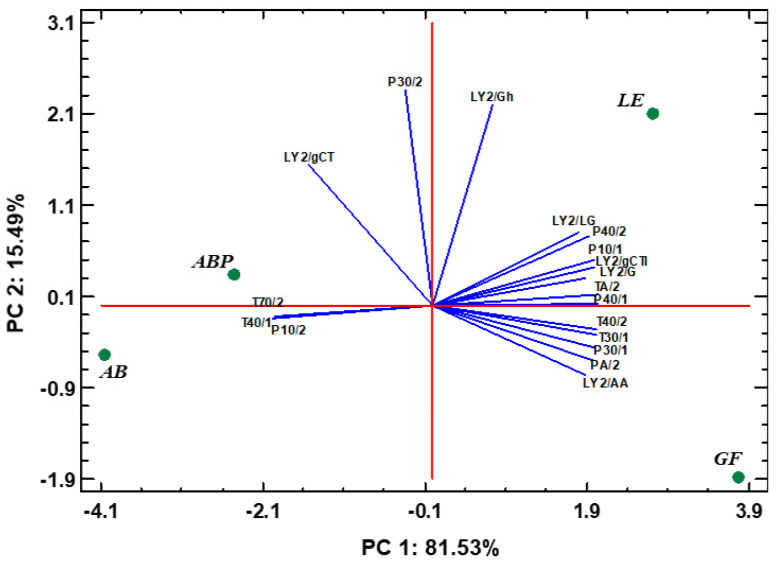
Multivariate analysis (principal components analysis) corresponding to the odor profile of the four different freeze-dried mushrooms evaluated by the response intensity of 18 sensors (variables) from an 𝛼-FOX electronic nose (AB, *Agaricus bisporus* sp. *bisporus*; ABP, *Agaricus bisporus* sp. *brunnescens*; LE, *Lentinula edodes*; GF, *Grifola frondosa*). Blue lines depict the intersection between the red axes, PC1 and PC2, and the PCs of the signal intensity detected by each sensor.

**Figure 3 jof-08-00953-f003:**
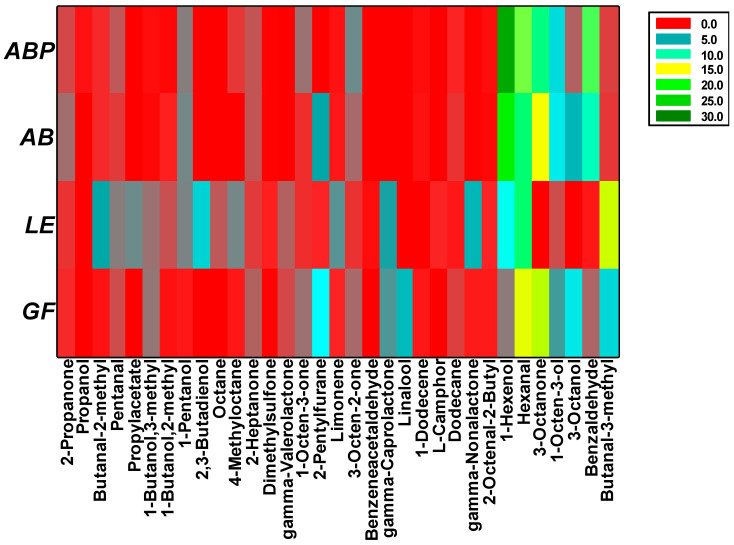
Percentage of most abundant VOCs, as detected by GC-MS, in the four freeze-dried samples under study (AB: *Agaricus bisporus* sp. *bisporus*; ABP: *Agaricus bisporus* sp. *brunnescens*; LE: *Lentinula edodes*; GF: *Grifola frondosa*).

**Figure 4 jof-08-00953-f004:**
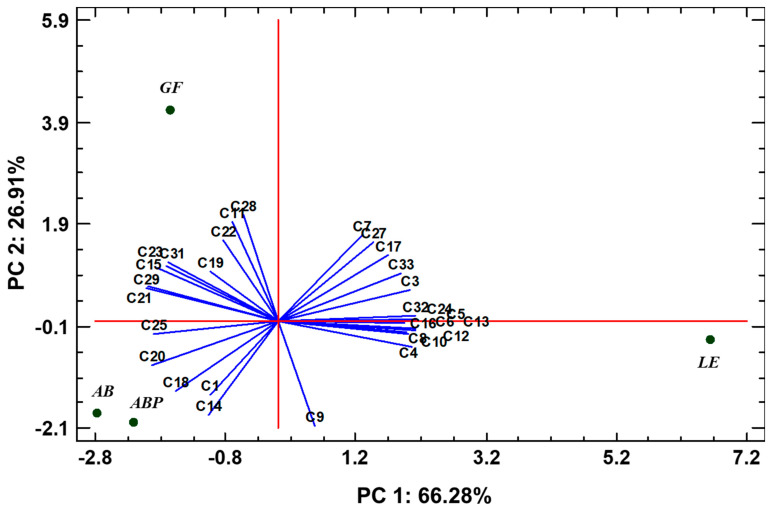
Multivariate analysis (principal component analysis) corresponding to the most-abundant volatile compounds in the freeze-dried mushroom samples detected by GC-MS (AB, *Agaricus bisporus* sp. *bisporus*; ABP, *Agaricus bisporus* sp. *brunnescens*; LE, *Lentinula edodes*; GF, *Grifola frondosa*). Most-abundant VOCs represented: C1: 2-Propanone; C3: Butanal-3-methyl; C4: Butanal-2-methyl; C5: Pentanal; C6: Propylacetate; C7: 1-Butanol,3-methyl; C8: 1-Butanol,2-methyl; C9: 1-Pentanol; C10: 2,3-Butadienol; C11: Hexanal; C12: Octane; C13: 4-Methyloctane; C14: 1-Hexenol; C15: 2-Heptanone; C16: Dimethylsulfone; C17: gamma-Valerolactone; C18: Benzaldehyde; C19: 1-Octen-3-one; C20: 1-Octen-3-ol; C21: 3-Octanone; C22: 2-Pentylfurane; C23: 3-Octanol; C24: Limonene; C25: 3-Octen-2-one; C27: gamma-Caprolactone; C28: Linalool; C29: 1-Dodecene; C31: Dodecane; C32: gamma-Nonalactone; and C33: 2-Octenal-2-Butyl. Blue lines depict the intersection between the red axes, PC1 and PC2, and the PCs of the different volatile compounds.

**Table 1 jof-08-00953-t001:** Combinations of the different series.

Series	Sample Combination					
Serie 1 (^α^ A vs. B)	AAB	ABA	ABB	BAA	BAB	BBA
Serie 2 (A vs. C)	AAC	ACA	ACC	CAA	CAC	CCA
Serie 3 (A vs. D)	AAD	ADA	ADD	DAA	DAD	DDA
Serie 4 (B vs. C)	BBC	BCB	BCC	CBB	CBC	CCB
Serie 5 (B vs. D)	BBD	BDB	BDD	DBB	DBD	DDB
Serie 6 (C vs. D)	CCD	CDC	CDD	DCC	DCD	DDC

^α^ A, *Agaricus bisporus* sp. *bisporus* (AB); B, *Agaricus bisporus* sp. *brunnescens* (ABP); C, *Lentinula edodes* (LE); D, *Grifola frondosa* (GF).

**Table 2 jof-08-00953-t002:** Instrumental color parameters (L*, a* and b*) of the freeze-dried mushroom samples under evaluation.

Species	Parameters		
	L*	a*	b*
^α^ AB	81.71 ± 1.70 a ^β^	0.41 ± 0.10 c	10.45 ± 0.46 c
ABP	81.14 ± 0.36 a	0.96 ± 0.06 b	9.00 ± 0.22 d
LE	79.37 ± 2.48 c	0.07 ± 0.06 d	17.94 ± 0.56 a
GF	78.63 ± 0.49 b	2.67 ± 0.19 a	12.38 ± 0.78 b

^α^ AB: *Agaricus bisporus* sp. *bisporus*; ABP: *Agaricus bisporus* sp. *brunnescens*; LE: *Lentinula edodes*; GF: *Grifola frondosa*. ^β^ Values followed by different letters within a column are significantly different at *p* ≤ 0.05 (LSD Fisher’s test).

**Table 3 jof-08-00953-t003:** Results of the sensory evaluation, by triangle test (54 panelists), of the series presented.

Series	Number of Correct Answers
Serie 1 (^α^ AB vs. ABP)	19
Serie 2 (AB vs. LE)	31
Serie 3 (AB vs. GF)	41
Serie 4 (ABP vs. LE)	28
Serie 5 (ABP vs. GF)	45
Serie 6 (LE vs. GF)	34

^α^ AB: *Agaricus bisporus* sp. *bisporus*; ABP: *Agaricus bisporus* sp. *brunnescens*; LE: *Lentinula edodes*; GF: *Grifola frondosa*.

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
