# Peer review of "Odor Profile of Four Cultivated and Freeze-Dried Edible Mushrooms by Using Sensory Panel, Electronic Nose and GC-MS"

_jof, 2022, doi:10.3390/jof8090953_

Round 1
Reviewer 1 Report
Please see the attachment

Author Response
Dear Reviewer,
Many thanks for your valuable comments and suggestions. The authors have now reviewed the document and implemented the corrections as corresponds. Beside, we have made an effort to reinforce the Introduction as suggested.
Reviewer 1:
General characterization of the manuscript (MS)
The authors have carried out a quantitative comparison between different methodologies strengthened by the chromatographic analysis to characterize a composition profile of nutritional products based on nutritional and medicinal mushrooms, that presents a prompt challenge.
Proposed corrections to manuscript
- The Introduction section should be extended to display the current achievements in the research area directly related to the tasks posed in MS, i.e. in the area of analysis of a composition profile of mushroom products, with the impact on (or even limited by) three genera under study in S: Agaricus, Lentinula and Grifola.
As suggested the following paragraph has been added:
Noteworthy, the odor profile of the cultivated genera Agaricus (showing high content of alcohols and aldehydes) [12], Lentinula (with sulfur containing compounds, eight-carbon compounds, and aldehyde compounds showing great contributions to mushroom flavor) [13] and Grifola (showing esters, alkanes, aldehydes and terpenes as major components) [14] has been analyzed by GC-MS. However, the mechanisms driving the odor profile of mushrooms by the substrate used or the drying process applied is not described and understood [12, 13].
- Few sentences in the Introduction should be dealt with the techniques used, their essence and advantages.
The introduction has been refurbished as suggested with the aim of presenting the techniques employed and the rationale of our approach to select the methodologies applied for the discrimination of samples. The three techniques are proved to be complementary and offer comparable results that reinforce the singular VOC profile of the mushroom samples tested as is presented. For this purpose, introductory paragraphs from the M&M have been recycled to conform part of the Introduction:
“Gas chromatography–mass spectrometry (GC-MS) combines gas-chromatography (for separation of compounds) and mass spectrometry (for the structural determination of compounds) to identify different substances within a volatile test sample [11]. Noteworthy, the odor profile of the cultivated genera Agaricus (showing high content of alcohols and aldehydes) [12], Lentinula (with sulfur containing compounds, eight-carbon compounds, and aldehyde compounds showing great contributions to mushroom flavor) [13] and Grifola (showing esters, alkanes, aldehydes and terpenes as major components) [14] has been analyzed by GC-MS. However, the mechanisms driving the odor profile of mushrooms by the substrate used or the drying process applied is not described and understood [12, 13].
In addition to the GC-MS, the odor profile of fresh and dried samples of cultivated button mushroom (Agaricus bisporus) has been evaluated by using electronic nose with comparable results [12]. The electric nose is an analytical device that permits automatic discrimination among different samples based on the odor profile (composition of their volatile fraction). In this device, various sensors exhibit a particular selectivity and sensitivity with respect to singular components of the sample, eventually generating a “fingerprint” of the chemical image in the volatile mixture [15].
Although analytical devices can prevent subjective mistakes, humans are able to detect a broad range of odors since the human olfactory epithelium contains ≈106–107 total olfactory receptor neurons [16]. The triangle sensory tests are discriminatory tests that aim to establish differences among diverse samples instead of evaluating subjective sensation experienced by the tested individuals [17]. Triangular tests have been successfully employed for discrimination of samples from wild edible and toxic mushroom of the Amanita genus in combination with electronic nose [18].”
- Novelty of the MS should be clearly pointed out in the Introduction. For instance, from the Lines 418-421, which contain the important information regarding the substrate formulation's weak (if any) impact on the odor or taste profile of mushrooms. Novel aspects of the triangle test use in this work should also be mentioned.
The noted paragraph has been modified and extended as suggested. And new references have been included, see comments to reviewer 3 below.
Other comments:
- Line 72-78: It would be better to rephrase this long sentence to yield several shorter ones.
The sentence has been modified as suggested: “For instance, cultivated edible and medicinal mushrooms such as Agaricus bisporus (button mushroom), and Lentinula edodes (shiitake) stand out for their high content in β-glucans [8]. Antioxidant substances/free-radical scavengers such as ascorbic acid, caffeic acid, adenosine, ergosterol, ergothioneine or glutathione have been detected in the fruiting body and mycelium of Hericium erinaceus (lion´s mane) [9]. Besides, mushroom bioactive peptides (MBAPs) with antioxidant and antimicrobial activities and effective to reduce blood pressure have been reported in Agaricus bisporus, Grifola frondosa (hen of the woods) or Letinula edodes [10].”
- Lines 105-106. This sentence would better be anywhere in Discussion.
This sentence introduce the methodology employed, and it refers to the writing style of the authors. We consider that this short introduction to the Materials and methods employed (in this case the biology of the organisms cultivated require the use of selective substrates for the growth of the mushrooms evaluated) contributes to the understanding of the methodology and facilitate reader’s comprehension. Hopefully, this explanation is sufficient to reconsider the reviewer´s recommendation.
- Line 117: "strain ST1 Lentinula edodes " is more familiar as "Lentinula edodes, strain ST1". Analogously, "Grifola frondosa, strain M9827".
Amended.
- Lines 178-184: These sentences would better be anywhere in Discussion, or may be in Introduction to follow point 2.
This sentence introduce the methodology employed, but as suggested have been moved to the Introduction.
- Line 227: "0255 µm film thickness" should be replaced by "0.255 µm film thickness" (or may be 0.25 µm?).
Corrected: 0.25 µm
- Lines 398-400: A component of the sentence "...suggesting that triangle test is limiting to discriminate between varieties of the same species but not among species belonging to the same or different genera ..." could be read as having the opposite sense, looking like "triangle test discriminates between only the same-species samples, but not between different species or genera" in spite of their much more profound distinctions.
The sentence has been refurbished: “…suggesting that triangle test is limiting to discriminate between varieties of the same species but permits differentiation among different species belonging to the same genera”
Yours sincerely,
Jaime Carrasco, PhD.
Reviewer 2 Report
The article presents information of interest; however, it is not clear to me to what degree it can be used in commercial products. I send some comments:
1) The authors refer to the importance of their work to avoid fraud in mushroom products but only make the comparison of species that were cultivated by them, so the management parameters were similar, which ensures that there is no effect of other parameters so that they will perform the comparison of the tested methods. However, I think they should have purchased commercial products (mushroom powder) to compare, the profile reported by the authors, so it would be more realistic than if you can discriminate between products that do not contain mushroom powder.
2) I suggest reviewing what they indicate in lines 417-418, as there are reports indicating the effect of the substrate on the organoleptic characteristics of fungi. The work: Li, W. et al. (2019). Analysis of volatile compounds of Lentinula edodes grown in different culture substrate formulations. Food research international, 125, 108517, reports that there was an effect on the profile of volatile compounds depending on the type of substrate used. In the work reported by: Sun et al. (2020). Advances in umami taste and aroma of edible mushrooms. Trends in Food Science & Technology, 96, 176-187, indicate that there are differences in the taste and aroma of umami between cultivated and wild mushrooms. In addition, we must also consider the type of processing (harvesting, drying, grinding, etc.) that can also modify the characteristics, the work reported by: Du, X. et al. (2021). Aroma and flavor profile of raw and roasted Agaricus bisporus mushrooms using a panel trained with aroma chemicals. LWT, 138, 110596, refers to temperature having an impact on the perceived sensory quality of the product, taste and texture characteristics.
3) In the same text (lines 417-418) they indicate that if there is an effect on performance and quality, they should be more specific than they refer to quality since there are authors who within the quality includes taste and smell.
Author Response
Dear Reviewer,
Many thanks for your valuable comments and suggestions. The authors have now reviewed the document and implemented the corrections as corresponds. Beside, we have made an effort to reinforce the Introduction as suggested.
Reviewer 2:
The article presents information of interest; however, it is not clear to me to what degree it can be used in commercial products. I send some comments:
1) The authors refer to the importance of their work to avoid fraud in mushroom products but only make the comparison of species that were cultivated by them, so the management parameters were similar, which ensures that there is no effect of other parameters so that they will perform the comparison of the tested methods. However, I think they should have purchased commercial products (mushroom powder) to compare, the profile reported by the authors, so it would be more realistic than if you can discriminate between products that do not contain mushroom powder.
We really appreciate the comments of the reviewer; however, the suggested approach does not reflect the rationale of our approach. In our study we have controlled the process of cultivation while using commercial substrates specifically designed for the commercial cultivation of the selected edible and medicinal mushrooms under study. The crop cycle has been controlled to achieve healthy carpophores. Therefore, while comparing commercial samples in the market and ours, we can lose sensitive information that can alter the results. Besides we have also controlled the process of dehydration by freeze-drying in our lab, it is very rare to find freeze-dried mushroom samples in the market since most of them are oven dried and the process of dehydration may alter the olfactory profile in the samples.
Comparison of cultivated mushrooms under standardized and controlled conditions of cultivation and postharvest treatment with commercial samples available in the market can be the object of future work but here we present differences among mushroom samples that have been cultivated and dehydrated by us in the lab and our research facilities for mushroom cultivation.
2) I suggest reviewing what they indicate in lines 417-418, as there are reports indicating the effect of the substrate on the organoleptic characteristics of fungi. The work: Li, W. et al. (2019). Analysis of volatile compounds of Lentinula edodes grown in different culture substrate formulations. Food research international, 125, 108517, reports that there was an effect on the profile of volatile compounds depending on the type of substrate used. In the work reported by: Sun et al. (2020). Advances in umami taste and aroma of edible mushrooms. Trends in Food Science & Technology, 96, 176-187, indicate that there are differences in the taste and aroma of umami between cultivated and wild mushrooms. In addition, we must also consider the type of processing (harvesting, drying, grinding, etc.) that can also modify the characteristics, the work reported by: Du, X. et al. (2021). Aroma and flavor profile of raw and roasted Agaricus bisporus mushrooms using a panel trained with aroma chemicals. LWT, 138, 110596, refers to temperature having an impact on the perceived sensory quality of the product, taste and texture characteristics.
As suggested by the reviewer the following paragraph and references have been added to discussion:
“Likewise, a brief number of reports have evaluated the effect of the substrate on the organoleptic characteristics of cultivated mushrooms [13, 42]. The relationship between volatile compounds production and culture substrates during Lentinula edodes grown in different substrates formulations was evaluated [13]. The authors conclude that the carbon source for shiitake cultivation can be partially modified by using bagasse instead of sawdust. However, the mechanisms driving the volatile composition as a result of the carbon source used for substrate formulation require further research [13]. Besides, variations on umami taste and aroma (considered an important factor for consumer choice) in cultivated mushrooms have been also correlated to the use of different substrates [42]. However, as noted by the authors, mechanisms underlying the variations in umami components of different mushrooms are likewise unclear. Therefore, to date the impact of substrate formulation as agronomical trait configuring the odor or flavor profile is not well understood and can be the object of future research.”
In respect to the type of processing, this aspect has been discussed, the following sentence and reference has been added in line 451:
“The different observations reported suggest that oven drying in comparison to freeze-drying can provide different bad distinctive volatile profile to the dried mushroom samples obtained after both procedures. This hypothesis is also reinforced by the results of Du et al. [45] who reported significant sensory differences among three different strains of A. bisporus, sharper by roasting process.”
New reference: 37.41. Du, X.; Sissons, J.; Shanks, M.; Plotto, A. Aroma and flavor profile of raw and roasted Agaricus bisporus mush-rooms using a panel trained with aroma chemicals. LWT 2021, 138, 110596, doi: 10.1016/j.lwt.2020.110596.
3) In the same text (lines 417-418) they indicate that if there is an effect on performance and quality, they should be more specific than they refer to quality since there are authors who within the quality includes taste and smell.
The following sentence has been modified as suggested, and a new reference has been included:
“The substrate formulation has significant impact on mushroom parameters such as biological efficiency, crop earliness, unitary weight, mushroom composition, commercial class and firmness”
New reference: “Pardo-Giménez, A.; Catalán, L.; Carrasco, J.; Álvarez-Ortí, M.; Zied, D.; Pardo, J. Effect of supplementing crop substrate with defatted pistachio meal on Agaricus bisporus and Pleurotus ostreatus production. J. Sci. Food Agric. 2016, 96, 3838–3845. doi: 10.1002/jsfa.7579”.
Yours sincerely,
Jaime Carrasco, PhD.
Reviewer 3 Report
Regarding the manuscript entitled "Odor Profile of Four Cultivated and Freeze-dried Edible Mushrooms by Using Sensory Panel, Electronic Nose and GC-MS", I have some recommendations in order to enhance its quality. As general comments, please read and check the language carefully again, please find below my comments:
L61: please cite proper reference for “and just around 30 species are cultivated commercially”
L150: I do not understand meaning “Humans are able to detect a broad range of odors that modify human behavior”
L218-221: no need to describe “GC-MS”, as well as “Solid phase microextraction (SPME)” (L206)
L224: specify please features of the used “mass spectrometer detector”
Section 3.4. please specify values with “respectively”
In “Discussion” section, please mention in brief major biological activities of the mushrooms’ essential oils/volatile constituents, if available, otherwise the extracts effects
Author Response
Dear Reviewers,
Many thanks for your valuable comments and suggestions. The authors have now reviewed the document and implemented the corrections as corresponds. Beside, we have made an effort to reinforce the Introduction as suggested.
Reviewer 3:
Comments and Suggestions for Authors
Regarding the manuscript entitled "Odor Profile of Four Cultivated and Freeze-dried Edible Mushrooms by Using Sensory Panel, Electronic Nose and GC-MS", I have some recommendations in order to enhance its quality. As general comments, please read and check the language carefully again, please find below my comments:
L61: please cite proper reference for “and just around 30 species are cultivated commercially”
The cited sentence has been modified as follows:
“…and just around 12 species are cultivated for commercial use as food or medicine [Ferraro et al., 2020].”
And the reference has been added:
“Ferraro, V.; Venturella, G., Pecoraro, L.; Gao, W.; Gargano, M. L. Cultivated mushrooms: importance of a multi-purpose crop, with special focus on Italian fungiculture. Plant Biosyst 2020, 1-11, doi: 10.1080/11263504.2020.1837283.”
L150: I do not understand meaning “Humans are able to detect a broad range of odors that modify human behavior”
As suggested, this introductory paragraph has been modified as:
“Humans are able to detect a broad range of odors since the human olfactory epithelium contains ≈106–107 total olfactory receptor neurons (Doleman and Lewis, 2001).”
The following reference has been added: “16. Doleman, B. J.; Lewis, N. S. Comparison of odor detection thresholds and odor discriminablities of a conducting polymer composite electronic nose versus mammalian olfaction. Sens Actuators B Chem 2001, 72(1), 41-50, doi: 10.1016/S0925-4005(00)00635-3.”
The following sentence and reference have been eliminated:
“Humans are able to detect a broad range of odors that modify human behaviour [16].”
- Oleszkiewicz, A.; Heyne, L.; Sienkiewicz-Oleszkiewicz, B.; Cuevas, M.; Haehner, A.; Hummel, T. Odours count: human olfactory ecology appears to be helpful in the improvement of the sense of smell. Sci Rep 2021, 11(1), 1-10, doi: 10.1038/s41598-021-96334-3.
L218-221: no need to describe “GC-MS”, as well as “Solid phase microextraction (SPME)” (L206)
The sentence was wrongly expressed. Authors have modified it:
“The VOCs profile of different mushroom species was analyzed by a static Agilent 6890N system GC-MS-FID (Termoquest, Milan, Italy).”
Sentence in L206 has been eliminated together with the reference cited: “
“Solid phase microextraction (SPME) is a versatile and reliable method for the rapid sampling, isolation, and analysis of the volatile fraction of a particular biological matrix [22].”
L224: specify please features of the used “mass spectrometer detector”
The features of the detector have been added: “… GC-MS-FID (Termoquest, Milan, Italy)”, as broadly known by the scientific community, FID is short for “flame ionization detector”.
Section 3.4. please specify values with “respectively”
L328-L332: The paragraph has been modified to specify the species that content the corresponding VOC in the percentage detected for AB or ABP:
“...1-hexenol (23.33 % in AB and 27.75 % in ABP), and hexanal (11.42 % in AB and 12.41 % in ABP), that are described as fruity aroma [25], the close related pair 3-octanone (14.72 % in AB and 10.99 % in ABP) and 1-octen-3-ol (6.76 % in AB and 6.30 % in ABP), secondary metabolites present in most mushrooms and responsible of their typical “mushroom-like” odor [26], and benzaldehyde (11.96 % in AB and 9.34% in ABP),…”
In “Discussion” section, please mention in brief major biological activities of the mushrooms’ essential oils/volatile constituents, if available, otherwise the extracts effects
This section is already included in the introduction and the discussion, since we do not aim to discuss the biological profile of the VOCs analyzed/extracts from mushrooms in depth but the singularities of the samples as fingerprint for the different samples. We consider that this is sufficiently covered and adding further information can provoke redundancy for the reader:
L452: “Lentinula edodes (shiitake) and Grifola frondosa (maitake) have been deeply analyzed for their medicinal value. Among their active principles with multiple beneficial uses, for instance the polysacharide lentinan from LE [46] and grifon from GF [47] posses immunemodulating and antitumor effects.”
They have been also included in the introduction: “Some species are appreciated due to their broad spectrum medicinal and pharmacological properties. Among the health benefits reported, cultivated medicinal mushrooms exhibit antibiotic, cytotoxic, immunomodulating, anti-inflammatory, antioxidative, antiallergic, antidepressive, antihyperlipidemic, antidiabetic, digestive, hepatoprotective, neuroprotective, nephroprotective, osteoprotective, and hypotensive activities [6]. Many different bioactive metabolites have been extracted and characterized from cultivated medicinal mushrooms, including carbohydrates, sterols, sphingolipids, fatty acids, sesquiterpenes, peptides or enzymes [7]. For instance, cultivated edible and medicinal mushrooms such as Agaricus bisporus (button mushroom), and Lentinula edodes (shiitake) stand out for their high content in β-glucans [8]. Antioxidant substances/free-radical scavengers such as ascorbic acid, caffeic acid, adenosine, ergosterol, ergothioneine or glutathione have been detected in the fruiting body and mycelium of Hericium erinaceus (lion´s mane) [9]. Besides, mushroom bioactive peptides (MBAPs) with antioxidant and antimicrobial activities and effective to reduce blood pressure have been reported in Agaricus bisporus, Grifola frondosa (hen of the woods) or Letinula edodes [10].”
Yours sincerely,
Jaime Carrasco, PhD